# SymboLLM-FE: LLM Accelerated Symbolic Regression
# for Automated Feature Engineering

**Zi-Jian Cheng** [1 2]  **Zi-Yi Jia** [1 2]  **Zhi Zhou** [2 3]  **Yu-Feng Li** [2 3]  **Lan-Zhe Guo** [1 2]

## Abstract

Tabular data, as a core data format in machine learning, often lacks the discriminative power needed for high-performance modeling due to insufficient feature informativeness. Automated Feature Engineering (AutoFE) overcomes this by automating feature generation and selection, ensuring both model performance and operational efficiency. However, traditional AutoFE often yield features with poor interpretability because they rely on blind mathematical transformations, while large language models (LLM)-based AutoFE faces challenges in requiring costly multi-round iterations to generate high-utility features to effectively enhance model performance, compounded by inherent risks of bias and hallucination. In this paper, we combine **symbo**lic regression with **LLM**s for **f**eature **e**ngineering (SymboLLM-FE) to solve these challenges. We extract mathematically expressive formulas strongly correlated with the target via symbolic regression, which can enhance model performance, then refine them by LLMs with rich prior knowledge to ensure interpretability. Empirical results on six real-world datasets and four Kaggle competitions demonstrate that SymboLLM-FE outperforms existing AutoFE. SymboLLM-FE also addresses the dual challenges of poor interpretability and numerous iterations by employing a statistical prior-grounded LLM refinement mechanism and single-digit LLM calls.

[1]School of Intelligence Science and Technology, Nanjing University, China [2]National Key Laboratory for Novel Software Technology, Nanjing University, China [3]School of Artificial Intelligence, Nanjing University, China. Correspondence to: Lan-Zhe Guo <guolz@nju.edu.cn>.

*Proceedings of the $2^{nd}$ ICML Workshop on Foundation Models for Structured Data*, Seoul, South Korea. 2026. Copyright 2026 by the author(s).

## 1. Introduction

Tabular data (Altman & Krzywinski, 2017), characterized by its structured row-column organization, is pivotal in domains ranging from finance (West, 2000; Zhu et al., 2021) to healthcare (Yıldız & Kalayci, 2024; Meijerink et al., 2020). However, its utility is often constrained by insufficient feature informativeness and implicit high-order interactions, which hinder models from capturing underlying structures and lead to suboptimal performance (Sayed et al., 2025; Gorishniy et al., 2021). To mitigate these limitations, feature engineering is employed to enhance the mapping between inputs and targets (Ravishankar & Battineni, 2025).

Given the prohibitive labor costs of manual feature construction (Susan & Tuteja, 2024), AutoFE has emerged as a critical research direction. Traditional AutoFE approaches, such as beam search-based hybrid optimization (Horn et al., 2019), dynamic pruning (Zhang et al., 2023), and feature selection (Li et al., 2017; Chandrashekar & Sahin, 2014), rely on predefined transformations or exhaustive strategies to construct features. Despite their efficacy in boosting predictive accuracy, these methods suffer from poor interpretability. The generated features often lack explicit semantic meaning, creating a disconnect from domain-specific logic. This opacity is particularly detrimental in high-stakes applications like factor mining (Wang et al., 2025; 2026), where interpretability is essential for validating economic rationale, ensuring regulatory compliance and distinguishing robust patterns from spurious correlations.

Recent advancements in LLMs have revolutionized feature engineering by leveraging their capabilities in semantic understanding (Hollmann et al., 2024), logical reasoning (Zou et al., 2026), and knowledge guidance (Nam et al., 2024). Unlike traditional AutoFE methods constrained by predefined rules or exhaustive search, LLM-based approaches utilize rich prior knowledge and contextual comprehension to identify latent relationships within raw tabular data. This enables the construction of semantically meaningful and interpretable features through mathematical and logical operations. However, LLM-based AutoFE faces two critical challenges. Firstly, the necessity for extensive iterative validation with downstream models incurs prohibitive computational overhead and limits scalability (Abhyankar et al.,

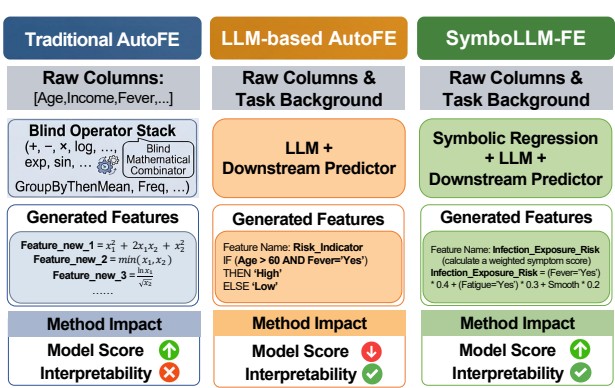

*Figure 1.* Comparative analysis of feature generation mechanisms and interpretability paradigms across AutoFE.

*Table 1.* Comparative analysis of performance efficiency and computational cost across AutoFE. AutoFeat and OpenFE are traditional AutoFE which don't rely on LLMs. LLM-RANK does not generate features because it's a feature selector.

| AutoFE | Generated Features | Score | API Call |
|---|---|---|---|
| AutoFeat | 2 | 79.75 | – |
| OpenFE | 1436 | 79.53 | – |
| CAAFE | 30 | 79.94 | 10 |
| OcTree | 50 | 79.52 | 50 |
| FEBP | 200 | 79.36 | 29 |
| LLM-FE | 30 | 79.59 | 35 |
| LLM-RANK | – | 78.86 | 7 |
| SymboLLM-FE | 70 | 80.02 | 4 |

- SymboLLM-FE addresses the dual challenges of poor interpretability and iterative experimentation by employing a statistical prior-grounded LLM refinement mechanism and single-digit LLM calls.

## 2. Problem Formulation and Analysis

### 2.1. Problem Formulation

Formally, the goal of tabular prediction tasks is to train a machine learning model $f : \mathcal{X} \rightarrow \mathcal{Y}$, where $\mathcal{X}$ is the input space and $\mathcal{Y}$ is the output space. We define the set of features in $\mathcal{X}$ as $C$. For the $n$-dimensional $\mathcal{X}$, the ordinary feature set $C = \{c_1, c_2, ..., c_n\}$.

Feature engineering is aimed to expand the dimension of $\mathcal{X}$ through generating $m$ new features, namely, the new feature set $C_{new} = \{c_{n+1}, c_{n+2}, ..., c_{n+m}\}$ based on $C$. After feature engineering, the dimension of $\mathcal{X}$ increases from $n$ to $n + m$ and the performance $\mathcal{P}$ of $f$ can be improved on the new feature set $\mathcal{S}$. Our aim is to get the maximum of performance $f$, by changing $C_{new}$ generated by $g$, i.e.,

$$\begin{aligned} \max \mathcal{P}_f(\mathcal{S}) &= \max \mathcal{P}_f(C \oplus C_{new}) \\ &= \max \mathcal{P}_f(C \oplus g(C)) \end{aligned} \quad (1)$$

### 2.2. Analysis

The comparative analysis of various AutoFE in Figure 1 and Table 1 reveals a distinct dichotomy between traditional and LLM-based AutoFE, highlighting critical bottlenecks in both efficiency and feature quality.

As illustrated in Figure 1, traditional AutoFE is strictly limited by uninterpretable features. Because it relies on a blind operator stack to mathematically combine raw columns without semantic understanding, it generates an excessively large feature set. These newly generated features lack semantic grounding and physical meaning, failing to provide actionable insights for future feature acquisition and offering poor

2025). Secondly, the inherent susceptibility of LLMs to hallucinations and biases may result in semantically invalid or unfair features, thereby compromising the reliability and predictive integrity of downstream models (Han et al., 2024).

It is evident that traditional AutoFE is limited by uninterpretable features, while LLM-based AutoFE struggles with the high iteration costs, hallucinations and implicit bias. Hence, it is urgent to research AutoFE for tabular machine learning tasks to develop high-efficiency feature engineering frameworks that yield interpretable features with significant performance gains to overcome limitations of paradigms.

To this end, we propose **SymboLLM-FE**, a framework synergizing symbolic regression and LLMs for interpretable feature engineering. It employs a Spearman-guided *expanding-sliding window* strategy to reduce search complexity from $\mathcal{O}(2^n)$ to $\mathcal{O}(n^2)$, enabling efficient derivation of mathematical candidates. Subsequently, LLMs inject domain priors via chain-of-thought reasoning, transforming opaque formulas into semantically meaningful code. This iterative process balances predictive performance with business-logic alignment, ensuring both accuracy and transparency.

Our contributions are summarized as follows:

- We conduct empirical analysis and identify that traditional AutoFE suffer from insufficient interpretability, while LLM-based AutoFE face extensive iterative experimentation, hallucination and implicit bias.

- We propose SymboLLM-FE to address current AutoFE challenges through leveraging symbolic regression to mine performance-enhanced features, which are then refined by LLMs to enhance interpretability.

- Comprehensive evaluations on six real-world datasets and four Kaggle competitions demonstrate the superior performance and generalizability of SymboLLM-FE compared with current AutoFE methods.

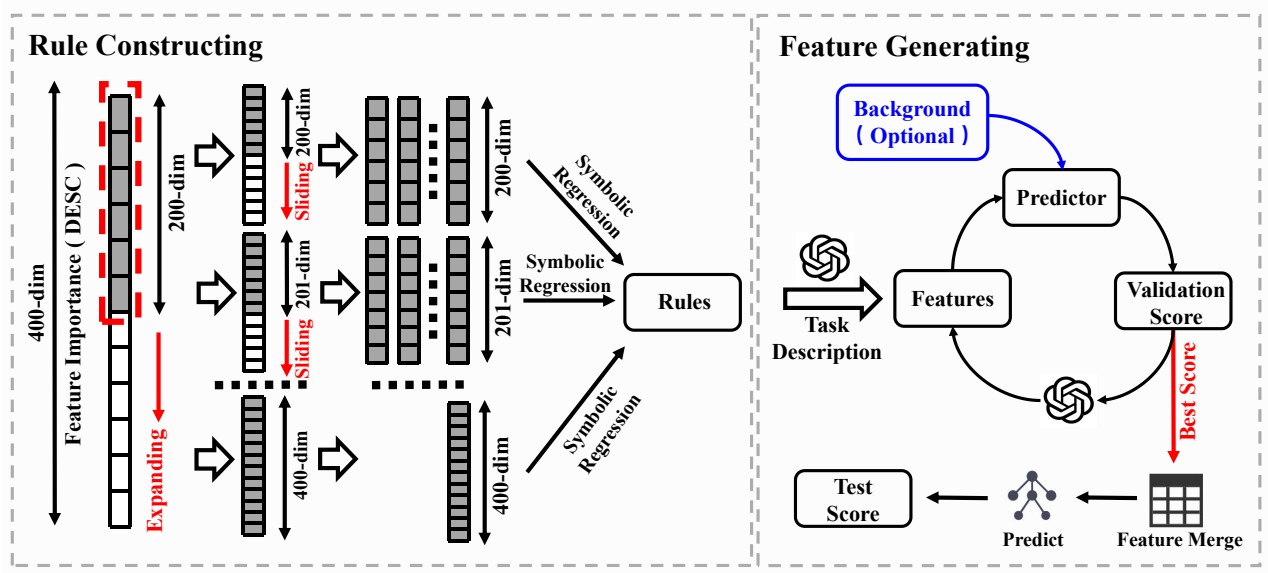

*Figure 2.* Overview of SymboLM-FE.

generalization to downstream tasks due to a severe compromise in the interpretability of generated features.

While LLM-based AutoFE ensures semantic interpretability by leveraging task background knowledge, it struggles with high iteration costs as shown in Table 1. Relying solely on LLMs for feature generation frequently yields features that, though semantically coherent and domain-relevant, lack robust discriminative power. Hence, methods like OcTree require approximately 50 rounds to converge, indicating that features generated by LLM-based AutoFE necessitate multiple rounds of trial-and-error validation and continuous refinement to enhance downstream model performance. This intrinsic uncertainty forces extensive iterative interactions with downstream predictors, leading to a prohibitive surge in both iteration rounds and computational time costs. Furthermore, these advanced LLM-based AutoFE remain inherently susceptible to severe hallucinations and biases, which can ultimately cause a drop in model performance.

These findings distinctly highlight the core limitations of existing traditional and LLM-based AutoFE, emphasizing the urgent need for a more optimized AutoFE framework that can bypass the high iteration overhead of LLMs and the uninterpretable nature of traditional mathematical combinations to generate features that are simultaneously interpretable and predictively effective.

## 3. Method

SymboLM-FE comprises four principal phases: (*i*) dynamic feature reordering via Spearman's rank correlation to prioritise target-relevant variables; (*ii*) application of symbolic regression through an expanding-sliding window mechanism to derive interpretable mathematical transformations; (*iii*) exploitation of LLMs' inductive reasoning capabilities for rule integration and optimisation, yielding executable code for novel feature construction; and (*iv*) concatenation of engineered features with the original feature space for downstream predictor validation, with iterative feedback of performance metrics, selected features, and corresponding codes into the LLMs until convergence to a relatively optimal solution. Figure 2 provides a detailed schematic illustration of the SymboLM-FE framework. Detailed descriptions of SymboLM-FE are shown in Appendix B.

### 3.1. Rule Construction by Symbolic Regression

We propose an expanding-sliding window approach based on Spearman feature importance to reduce search space complexity from traditional AutoFE.

**1. Feature Importance Ranking.** Calculate Spearman correlation $\phi$ for each feature $c_i \in C$ and sort them to obtain an ordered sequence $C'$:

$$C' = \{c'_1, c'_2, \ldots, c'_n\}, \quad \text{s.t. } \phi_{c'_1} \leq \phi_{c'_2} \leq \cdots \leq \phi_{c'_n} \quad (2)$$

**2. Expanding-Sliding Window.** Subsets $S$ are generated using a dynamic window of size $k$ ($k \in [\lfloor n/2 \rfloor, n]$): 1) Expanding: $k$ increases from $\lfloor n/2 \rfloor$ to $n$ with unit step. 2) Sliding: For each $k$, the window slides rightward. The total subset count $|\mathcal{S}|$ is $\sum_{k=1}^{n}(n-k+1)$.

**3. Rule Repository.** For each subset $\mathcal{S}_i \in S$, symbolic regression derives a rule $\tau_i$. The repository $\mathcal{R}$ is defined as:

$$\mathcal{R} = \{(\mathcal{S}_i, \mathcal{P}_f(\mathcal{S}_i), \tau_i)\}_{i=1}^{|\mathcal{S}|} \quad (3)$$

*Table 2.* Comparison of SymboLLM-FE with other AutoFE on real-world datasets. The best result is **bold** and the second best is underlined. The downstream model for prediction is TabPFN. '↑' means classification on accuracy and '↓' means regression on RMSE.

| FE Method | Credit-g ↑ | Spaceship ↑ | Cmc ↑ | Academic ↑ | Ailerons ↓ | Tesla ↓ |
|---|---|---|---|---|---|---|
| Baseline | 77.03±0.47 | 80.79±1.27 | 57.85±0.89 | 77.33±0.67 | 5.10±0.48 | 2.39±0.09 |
| AutoFeat | 77.83±1.43 | 80.76±1.17 | 57.85±0.32 | 76.80±0.46 | 5.04±0.45 | 2.38±0.00 |
| OpenFE | 76.50±3.34 | 80.30±1.00 | 57.85±2.35 | 76.42±0.61 | 5.26±0.49 | 2.18±0.06 |
| CAAFE | **78.00±0.71** | 80.99±1.00 | 57.78±1.28 | 77.29±0.56 | 5.03±0.46 | 2.66±0.09 |
| OcTree | 76.50±0.82 | 80.79±1.15 | 57.85±1.52 | 77.33±0.70 | 5.09±0.47 | 2.43±0.10 |
| FEBP | 77.50±0.41 | 80.85±1.29 | 57.93±0.48 | 77.36±0.35 | 5.08±0.48 | 2.41±0.08 |
| LLM-FE | 76.67±0.62 | 80.22±0.96 | 57.29±2.20 | 76.42±0.61 | 5.05±0.46 | 2.39±0.09 |
| LLM-RANK | 77.50±0.82 | 80.70±1.15 | 57.74±0.85 | 77.25±0.19 | 5.10±0.48 | 5.87±0.19 |
| SymboLLM-FE | 77.00±1.63 | **81.27±1.31** | **57.97±0.73** | **77.89±0.23** | **5.02±0.46** | **2.16±0.06** |

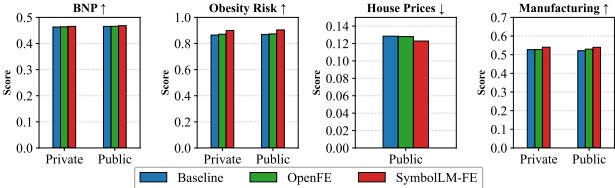

*Figure 3.* Score comparison on Kaggle competitions. '↑' means classification and '↓' means regression.

where $\mathcal{P}_f(\mathcal{S}_i)$ is the performance of symbolic model $f$ on subset $\mathcal{S}_i$.

## 3.2. Feature Generating by LLMs

We then implement a multi-stage pipeline to transform symbolic rules into executable code via LLMs.

- **LLM-guided Generation**: Rules $\tau_i \in \mathcal{R}$ and background knowledge are converted into natural language texts. LLMs generate Python/Pandas code via Chain-of-Thought prompt.

- **Performance Validation**: Fenerated codes are executed to produce an augmented feature set and enrich the original dataset. Then the enriched dataset is evaluated by a downstream predictor to obtain scores.

- **Iterative Refinement**: A closed-loop feedback mechanism is established:

$$\text{Feedback}_t = \{\text{Code}_t, \text{Score}_t, \text{LLM History}_{1:t-1}\} \quad (4)$$

The LLM optimizes the feature generation logic iteratively until performance converges.

## 4. Experiments

### 4.1. Main Results

Table 2 shows that SymboLLM-FE achieves statistically significant improvements over traditional AutoFE (average

gain of 1.23%) and approximately 1% higher accuracy than LLM-based AutoFE. Extensive experiments on CatBoost, XGBoost and MLP in Appendix G consistently demonstrate SymboLLM-FE's stable performance advantages across different model architectures. To assess model performance in the real world, we evaluate SymboLLM-FE under four Kaggle competitions. As shown in Figure 3, SymboLLM-FE+TabPFN consistently outperforms vanilla TabPFN and OpenFE+TabPFN, achieving an average score improvement of 2.5pp.

### 4.2. Analysis of Generated Features

As illustrated in Figure 1, SymboLLM-FE introduces a novel paradigm by incorporating symbolic regression alongside an LLM and a downstream predictor. Instead of relying blindly on mathematical combinations or unguided semantic spaces, SymboLLM-FE utilizes the LLM as a refinement engine rather than a primary generator. By grounding the LLM's input in $\tau_i$ and $\mathcal{P}$, we effectively anchor the feature generation process in statistical reality. This closed-loop framework ensures that the LLM's role is strictly limited to optimizing code implementation and integrating inductive logic based on prior knowledge. Consequently, it eliminates the generation of spurious or hallucinated features that lack a mathematical basis, yielding highly interpretable features such as weighted symptom scores (e.g., *Infection_Exposure_Risk* shown in Figure 1).

## 5. Conclusion

In this paper, we present SymboLLM-FE, a robust hybrid framework that combines symbolic regression's mathematical rigor with LLMs' refinement capabilities for AutoFE. SymboLLM-FE demonstrates consistent advantages over current AutoFE, enhanced interpretability and greater efficiency. Building on this, we further propose a reliable solution for real-world tasks by integrating SymboLLM-FE as a feature engineering module with TabPFN and show the effectiveness of this combined approach on Kaggle tasks.

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

## A. Related Work

### A.1. Tabular Machine Learning on LLMs.

The advent of LLMs possessing substantial domain-specific prior knowledge has prompted the exploration of their utility in addressing tabular machine learning tasks (Floridi & Chiriatti, 2020). For example, LIFT (Dinh et al., 2022) fine-tunes GPT-3 (Floridi & Chiriatti, 2020) on training data, showing that general-purpose LLMs can improve performance—though not surpassing tree-based models. TabLLM (Hegselmann et al., 2023) employs T0 (Sanh et al., 2021) for fine-tuning, effectively leveraging pre-trained knowledge with minimal labelled data. UniPredict (Wang et al., 2023) trains GPT-2 (Ethayarajh, 2019) across 169 tabular datasets, achieving competitive results without dataset-specific tuning. However, these LLM-based approaches demonstrate superior performance primarily in zero-shot or few-shot experimental settings (Hegselmann et al., 2023), while still lagging behind tree-based and deep learning models in terms of both computational efficiency and prediction accuracy in real-world scenarios (Cheng et al., 2025). This suggests that directly employing LLMs as predictors is suboptimal, prompting researchers to explore leveraging LLMs as feature engineering methods to enhance data representation, thereby improving the performance of downstream predictors like tree-based or deep learning models (Hollmann et al., 2024).

### A.2. Automated Feature Engineering on LLMs.

Automated feature engineering (AutoFE) supplants this artisanal paradigm with algorithmic search and data-driven optimization, systematically generating, selecting, and evaluating features. Recent advances in LLMs have driven progress in LLM-based AutoFE, which bifurcates into two key directions: feature generation and feature selection. For feature generation, methods include context-aware frameworks (Hollmann et al., 2024), decision tree for rule discovery (Nam et al., 2024), differentiable feature mapping (Han et al., 2024), structured generation via reverse Polish notation (Zou et al., 2026), and evolutionary-knowledge distillation hybrids (Abhyankar et al., 2025). Feature selection techniques employ log-probability thresholds (Choi et al., 2022) and ranking paradigms (Jeong et al., 2024). However, these approaches face challenges, including interpretability concerns and potential hallucinations and biases.

### A.3. Automated Feature Engineering on Symbolic Regression.

Symbolic Regression (Billard & Diday, 2002) is a machine learning method that discovers optimal mathematical expressions to uncover latent data relationships. Its capacity to derive interpretable analytical forms has made it valuable for feature engineering (Shmuel et al., 2024), improving both model transparency and predictive accuracy while reducing manual effort. Recent advances include evolutionary computation integration for physical phenomenon extraction (Xu et al., 2023), modular multi-tree genetic programming for high-dimensional feature reuse (San et al., 2021), and Spearman-based feature selection to enhance generalization (Kim et al., 2020). However, symbolic regression still struggles with expression complexity in high-dimensional spaces. While existing work has explored LLM-SR interactions for equation generation (e.g., natural language processing (Zhang et al., 2025) and code synthesis (Shojaee et al., 2025)), they have not extended its work to incorporate the equations generated by symbolic regression into LLMs for feature engineering.

## B. Method

To address the limitations of the current AutoFE, we shift our focus to synergize symbolic regression and LLMs. Symbolic regression excels in efficiently discovering explicit mathematical formulas that are not only computationally lightweight but also address the need for performance-enhancing features. Complementarily, LLMs leverage their knowledge to refine and optimize these candidate symbolic formulas. By injecting domain-specific priors, LLMs transform opaque mathematical formulas, simultaneously enhancing their interpretability through natural language explanations and ensuring their alignment with business logic. This hybrid paradigm, namely SymboLLM-FE, effectively bridges the gap between efficacy and interpretability. Figure 2 demonstrates the detailed pipeline of SymboLLM-FE.

**Dataset Split.** SymboLLM-FE strictly adheres to a fold-wise training protocol within a cross-validation framework. The training set is exclusively utilized for Formula Construction by Symbolic Regression, supporting the entire pipeline from Spearman correlation-based feature reordering and expanding-sliding window subset generation to the fitting of symbolic models. The validation set serves as the feedback environment for Feature Generation via LLMs. It remains isolated from the former process but is actively used to compute validation scores that guide the LLM's iterative code synthesis, rule mining, and feature optimization loop. Finally, the test set is reserved solely for the final downstream evaluation of the merged feature set to assess model performance.

## B.1. Formula Construction by Symbolic Regression

We firstly construct numerous input feature subsets by using Spearman correlation (De Winter et al., 2016) and an expanding-sliding window approach. Subsequently, symbolic regression model is applied to each input feature subset separately to derive interpretable mathematical formulas that capture significant relationships with the target.

**Feature Set Sampling.** Given $C$ with $n$ features, the total number of its non-empty feature subsets is $2^n - 1$. Applying symbolic regression to every subset would lead to exponential computational complexity ($\mathcal{O}(2^n)$). To optimize computational efficiency while preserving feature representational capacity, we propose a feature combination strategy based on Spearman correlation coefficient analysis and an expanding-sliding window approach.

Specifically, we first evaluate the importance of $C$ using Spearman correlation coefficient analysis, followed by a monotonic ascending sorting based on the correlation coefficients. This process yields an ordered feature sequence $C' = c'_1, c'_2, ..., c'_n$, where each feature satisfies the relation $\phi_{c'_1} \leq \phi_{c'_2} \leq ... \leq \phi_{c'_n}$, with $\phi_{c'_i}$ denoting the Spearman correlation coefficient between feature $c'_i$ and the target. We choose Spearman correlation as the criterion for feature importance quantification because, compared to other metrics such as Pearson correlation, Spearman correlation more robustly quantifies nonlinear dependencies between features and the target, thereby providing a more accurate measure of marginal feature importance (Cheng et al., 2025). Appendix F empirically validates the statistical correlations between this metric and other feature importance criteria.

To maintain formula quality while effectively controlling the quantity of generated features, we propose a dynamic expanding-sliding window approach. The approach initializes with window size $k = \lfloor n/2 \rfloor$ and operates through two core phases:

- **Expanding Phase**: $k$ incrementally increases from $\lfloor n/2 \rfloor$ to $n$ with unit step size.

- **Sliding Phase**: For each fixed $k$, the window slides rightward with single-feature steps to generate different feature subsets. For instance, when $k = \lfloor n/2 \rfloor$, the generated subsets are $\{\{c'_1, ..., c'_{n/2}\}, \{c'_2, ..., c'_{n/2+1}\}, ...\}$.

The total number $\mathcal{S}$ of candidate input feature subsets generated by the expanding-sliding window approach is upper bounded by an arithmetic series summation, i.e.,

$$\sum_{k=\lfloor n/2 \rfloor}^{n} (n - k + 1) = \sum_{m=1}^{\lceil n/2 \rceil + 1} m$$
$$= \begin{cases} \dfrac{(n+2)(n+4)}{8}, & \text{if } n \text{ is even} \\ \dfrac{(n+3)(n+5)}{8}, & \text{if } n \text{ is odd} \end{cases} \tag{5}$$

Equation 5 shows that this approach reduces the sum of feature subsets from exponential $\mathcal{O}(2^n)$ to polynomial $\mathcal{O}(n^2)$.

**Formula Generating.** For each $S_i$ where $i \in [1, |\mathcal{S}|]$ generated by the expanding-sliding window approach, we train one symbolic regression model and generate a formula like `add(mul(c1,c2),c3)` from $S_i$ to the target as the new feature. Appendix C shows details about symbolic regression. Through systematic training and construction, we ultimately establish a formula-based feature repository $\mathcal{R}$ containing $|\mathcal{S}|$ triples:

$$\mathcal{R} = \{(S_i, \mathcal{P}_f(S_i), \tau_i)\}_{i=1}^{|\mathcal{S}|} \tag{6}$$

where each triple consists of: 1) the feature subset $S_i$; 2) symbolic regression performance $\mathcal{P}$ of trained symbolic model $f$ on $S_i$; 3) the formula $\tau_i$ from $S_i$ to the target.

## B.2. Feature Generation via LLMs

We then implement a multi-stage refinement pipeline via LLMs and downstream predictors based on $\mathcal{R}$ to generate features.

*Table 3.* Hyperparameter Grids of Downstream Baselines.

| Model | Hyperparameter | Values |
|---|---|---|
| **XGBoost** | Learning Rate | $\{0.01, 0.1\}$ |
| | Max. Depth | $\{1, 5, 9\}$ |
| | N Estimators | $\{10000, 20000, 30000\}$ |
| | Subsample | $\{0.5, 0.8, 1.0\}$ |
| | Colsample Bytree | $\{0.5, 0.8, 1.0\}$ |
| | Min Child Weight | $\{1, 3, 5\}$ |
| **CatBoost** | Learning Rate | $\{0.01, 0.05, 0.1\}$ |
| | Depth | $\{4, 6, 8\}$ |
| | Iterations | $\{500, 1000, 2000\}$ |
| **MLP** | D_layers | $\{1, 8, 64, 512\}$ |
| | Dropout | Uniform $\{0.0, 0.5\}$ |
| | Learning Rate | Loguniform$\{e^{-5}, 0.01\}$ |
| | Weight Decay | Loguniform$\{e^{-6}, 0.001\}$ |

**LLM-guided Code Generation.** The input to LLMs is constructed by converting every formula $\tau$ in $\mathcal{R}$ into its corresponding natural language description, concatenated with structured dataset background information and code-generation prompt. Subsequently, LLMs generate executable feature engineering code in the format of Python pandas through chain-of-thought reasoning (Wei et al., 2022).

**Downstream Predictor Performance Validation.** The generated code is executed to produce an augmented feature set $C_{new}$ and then enrich the original dataset $C$, i.e., $C_{final} = C \oplus C_{new}$. Sequentially, we train a downstream predictor and get the model performance result under $C_{final}$.

**Iterative Feature Refinement.** The predictor performance, along with the utilized features and their corresponding generation codes, is then fed back into LLMs for iterative optimization and feature refinement. This feedback loop continues until the predictor achieves a relatively optimal performance level on the augmented feature set.

## C. Implementation Details of Symbolic Regression

This appendix section provides a comprehensive specification of symbolic regression employed within the SymboLLM-FE. We utilize the gplearn library (v0.4.1)[1], which implements a Genetic Programming paradigm. symbolic regression is designed to discover interpretable mathematical expressions by evolving a population of syntax trees, denoted as $\mathcal{T}$, through mechanisms inspired by natural selection.

### C.1. Algorithmic Framework

The core objective of symbolic regression is to search the space of mathematical functions to find an expression $\mathcal{T}^*$ that minimizes the prediction error on the training data. Let $\mathbf{X}_{train} \in \mathbb{R}^{N \times d}$ be the feature matrix and $\mathbf{y}_{train} \in \mathbb{R}^N$ be the target vector for a given subset. The Genetic Programming process proceeds through four iterative phases:

1. **Population Initialization:** An initial population $\mathcal{P}_0$ of size $P$ is generated. Each individual in $\mathcal{P}_0$ is a random syntax tree $\mathcal{T}_i$, where internal nodes represent function operators and leaf nodes represent input features or ephemeral constants.

2. **Fitness Evaluation:** For each individual $\mathcal{T}_i \in \mathcal{P}_t$ at generation $t$, we compute MSE on the training set as the raw fitness score $E(\mathcal{T}_i)$. To mitigate bloat, a parsimony penalty is added, yielding the composite fitness $\tilde{E}(\mathcal{T}_i)$:

$$\tilde{E}(\mathcal{T}_i) = E(\mathcal{T}_i) + \Omega \cdot \text{size}(\mathcal{T}_i) \tag{7}$$

where $\text{size}(\mathcal{T}_i)$ denotes the number of nodes in the syntax tree $\mathcal{T}_i$, and $\Omega$ is the parsimony coefficient controlling the trade-off between accuracy and complexity.

---

[1] https://github.com/trevorstephens/gplearn

*Table 4.* Symbolic regression operator set. Protected operations ensure numerical stability across the domain.

| Operator | Arity | Mathematical Form | Protection Strategy |
|---|---|---|---|
| add | 2 | $x + y$ | None |
| sub | 2 | $x - y$ | None |
| mul | 2 | $x \cdot y$ | None |
| div | 2 | $x/y$ | Returns 1 if $y = 0$ |
| sqrt | 1 | $\sqrt{\lvert x \rvert}$ | Absolute value input |
| inv | 1 | $1/x$ | Returns 0 if $x = 0$ |
| neg | 1 | $-x$ | None |
| abs | 1 | $\lvert x \rvert$ | None |
| max | 2 | $\max(x, y)$ | None |
| min | 2 | $\min(x, y)$ | None |
| sin | 1 | $\sin(x)$ | None |
| cos | 1 | $\cos(x)$ | None |
| tan | 1 | $\tan(x)$ | None |
| log | 1 | $\ln \lvert x \rvert$ | Returns 0 if $x = 0$ |

3. **Genetic Operations:** Parents are selected via tournament selection. New offspring are generated through:

   - *Subtree Crossover:* With probability $p_{crossover}$, subtrees from two parents are exchanged.
   - *Mutation:* With remaining probability, point mutation, subtree mutation, or reproduction is applied to introduce diversity.

4. **Iterative Evolution:** Steps 2 and 3 are repeated until the stopping criteria are met. The best individual $\mathcal{T}^*$ from the final population is returned as the derived symbolic rule.

For regression tasks, we employ the SymbolicRegressor. While for binary classification, the SymbolicClassifier is used.

### C.2. Operator Set Configuration

To balance expressive power with interpretability, we define a curated set of 14 primitive operators. This set covers arithmetic, transcendental, and piecewise nonlinear functions. Table 4 details the operator inventory.

The design rationale for primitive operators is threefold: (i) Basic arithmetic operators (add, sub, mul, div) form the backbone for rational function approximation; (ii) Transcendental functions (sin, cos, tan, log) enable the modeling of periodic and exponential dynamics; (iii) Piecewise operators (max, min, abs) allow for threshold-based logic. Crucially, all operators are *protected*, meaning they return safe default values (e.g., 1 or 0) when encountering undefined states (such as division by zero), ensuring that every syntax tree $\mathcal{T}$ is a valid function over $\mathbb{R}^d$.

### C.3. Hyperparameter Configuration

The hyperparameters for symbolic regression were fixed across all experiments to ensure consistency. The configuration, summarized in Table 5, was selected to maximize search coverage while maintaining computational feasibility.

A large population size ($P = 20,000$) combined with a high crossover probability ($p_{cx} = 0.9$) facilitates extensive exploration of the solution space. The tournament size of 100 imposes strong selection pressure, driving rapid convergence toward high-fitness regions.

### C.4. Regularization and Bloat Control

A prevalent challenge in GP is *bloat*, where the size of syntax trees $\mathcal{T}$ grows excessively without corresponding improvements in fitness, leading to overfitting and reduced interpretability. We address this via *parsimony pressure*, governed by the coefficient $\Omega$ in Eq. (1).

*Table 5.* Hyperparameter configuration for the symbolic regression.

| Parameter | Value | Rationale |
|---|---|---|
| Population Size ($P$) | 20,000 | Ensures genetic diversity in high-dimensional spaces. |
| Generations ($G_{max}$) | 120 | Hard upper bound on evolution steps. |
| Stopping Criteria ($\tau$) | $10^{-4}$ | Early stopping threshold for MSE. |
| Tournament Size | 100 | Moderate-to-high selection pressure. |
| Crossover Probability ($p_{cx}$) | 0.9 | Promotes recombination of successful sub-expressions. |
| Random State | 42 | Ensures reproducibility. |

The optimal value for $\Omega$ was determined via grid search over the candidate set $\{0.005, 0.01, 0.02, 0.03, 0.04, 0.05\}$ on a validation subset. The value minimizing the Root Mean Squared Error (RMSE) was selected for all subsequent experiments. This regularization ensures that the evolved expression $\mathcal{T}^*$ adheres to Occam's razor, favoring simpler structures when predictive performance is comparable.

## D. Prompt

In this section, we provide prompt templates for code generation.

## Prompt used for Code Generation

# Task Description
You are an expert data scientist tasked with improving a downstream classification model by generating new features and dropping redundant ones. The target variable is 'class'.

# Dataset Information
- The raw dataset has {N} columns named from 'X{0}' to 'X{N-1}'.
- The downstream task is {regression / classification}, and the evaluation metric is {accuracy / RMSE}.

# Prior Knowledge from Symbolic Regressor
The following table presents symbolic rules derived from the dataset. Each rule shows how the target variable can be approximated by combining selected feature subsets. The MAE indicates the performance of the Symbolic Regressor model on the test set using the corresponding formula.

$\|Formula\|MAE\|$
$\| - - - - - - \| - - - - - \|$
$\|\{rule_1\}\|\{mae_1\}\|$
$\|\{rule_2\}\|\{mae_2\}\|$
$\|...\|...\|$

# Instructions
Please consider these formulas and their performance comprehensively. Make full use of your prior knowledge to propose new features that can further improve the model's performance. Note that:
- New features can be directly derived from the formulas below, but you should **not** simply convert all formulas.
- Instead, you must analyze all formulas holistically and examine the relationships between the features.

# Code Requirements
You will write Python code to generate additional columns and optionally drop redundant columns. The code will be evaluated on a holdout set using accuracy.
## Naming Convention
- New column names must follow the existing naming scheme. - If the last existing column is 'X{k}', the first new column should be named 'X{k+1}', then 'X{k+2}', and so on.
## Format for Adding Columns
Each added column must include a comment block with feature name, reason, and usefulness.
```python
# Feature name: $new\_feature\_name$
# Reason: why this feature is proposed
# Usefulness: how it helps classify Class according to given rules
df['Xnext_index'] = ... # computation using existing columns
```

## Format for Dropping Columns
Each dropped column must include an explanation.
```python
# Explanation: why this column is redundant or harmful
df.drop(columns=['Xcol_index'], inplace=True)
```

## Code Block Rules
- Each code block starts with ' ```python ' and ends with ' ``` '.
- Added columns can be used in subsequent code blocks.
- Dropped columns are no longer available.

# Output
Generate one or more code blocks following the above format.

# E. Experimental Setup

## E.1. Datasets

We select a variety of open-source and reliable datasets from OpenML and Kaggle. These datasets encompass three primary tasks: binary classification, multi-class classification, and regression, and span a range of fields such as finance and healthcare. The information of used datasets are shown in Table 6.

**Credit-g.** Credit-g contains 1,000 entries with 20 categorical/symbolic attributes prepared by Prof. Hofmann. In this dataset, each entry represents a person who takes a credit from a bank. Each person is classified as having good or bad credit risk according to the set of attributes. The target is to determine whether the customer's credit is good or bad. This dataset is available at https://www.openml.org/search?type=data&sort=runs&id=31&status=active.

**Cmc.** Cmc is a supervised classification dataset predicting the contraceptive method used (three categories) based on married women's personal characteristics. It contains 1,473 samples and 9 attributes, commonly used for evaluating classification algorithms. This dataset is available at https://www.openml.org/search?type=data&sort=runs&id=23&status=active.

**Ailerons.** Ailerons is a regression dataset addressing an F16 aircraft control problem, predicting the control action on the ailerons based on the aircraft's status attributes. It contains 12,250 samples and 33 features, originating from a real aerospace control problem. This dataset is available at https://www.openml.org/search?type=data&sort=runs&id=296&status=active.

**Spaceship.** Spaceship is a beginner-level Kaggle competition dataset set in the year 2912, where a starship carrying nearly 13,000 passengers collides with a space-time anomaly, causing nearly half to be transported to another dimension. The dataset contains approximately 2,000 passenger records with features including home planet, cryo-sleep status, cabin number, destination, and spending records, predicting whether passengers were transported. This dataset is available at https://www.kaggle.com/competitions/spaceship-titanic.

**Academic.** The Academic dataset, sourced from Kaggle, is designed for predicting student outcomes. It contains demographic, socio-economic, and academic enrollment data used to classify students into categories such as "academic success" or "dropout." This dataset offers a more outcome-oriented perspective on student performance. This dataset is available at https://www.kaggle.com/datasets/missionjee/students-dropout-and-academic-success-dataset.

**Tesla.** The Tesla dataset, sourced from Kaggle, contains historical daily stock price data for Tesla Inc. (TSLA). Typical features include Open, High, Low, Close, Adjusted Close, and Volume. The dataset is commonly used for time series forecasting of stock price movements. This dataset is available at https://www.kaggle.com/datasets/guillemservera/tsla-stock-data.

*Table 6.* Datasets used in this paper.

| Dataset | Type | Samples | Features |
|---------|------|---------|----------|
| Credit-g | Binary Classification | 1,000 | 20 |
| Spaceship | Binary Classification | 2,000 | 13 |
| Cmc | Multi-class Classification | 1,473 | 9 |
| Academic | Multi-class Classification | 4424 | 36 |
| Ailerons | Regression | 12,250 | 33 |
| Tesla | Regression | 6906 | 8 |

## E.2. Downstream Baselines

To validate the generalization and adaptability of SymboLM-FE, we evaluate it on various tree-based models and deep learning models. For tree-based models, we choose CatBoost (Prokhorenkova et al., 2018) and XGBoost (Chen & Guestrin, 2016). For deep learning models, we choose MLP (Gorishniy et al., 2021) and TabPFN (Hollmann et al., 2025). We provide hyperparameter grids of tree-based and deep learning models in Table 3

**XGBoost.** XGBoost(Chen & Guestrin, 2016) is an efficient and flexible machine learning model that incrementally builds multiple decision trees by optimizing the loss function, with each tree correcting the errors of the previous one to continuously improve the model's predictive performance. XGBoost also incorporates the gradient boosting algorithm, iteratively training decision tree-based models with the goal of minimizing residuals and enhancing predictive accuracy.

*Table 7.* Comparison of SymboLLM-FE with other AutoFE on real-world datasets. The best result is **bold** and the second best is underlined. Downstream models for prediction are CatBoost, XGBoost, MLP and TabPFN. '↑' means classification on Accuracy and '↓' means regression on RMSE.

| Downstream Models | FE Methods | Credit-g ↑ | Spaceship ↑ | Cmc ↑ | Academic ↑ | Ailerons ↓ | Tesla ↓ |
|---|---|---|---|---|---|---|---|
| | Baseline | 75.83±3.52 | 80.41±0.26 | 56.50±2.63 | 77.02±0.42 | 5.13±0.56 | 3.84±0.10 |
| | AutoFeat | 77.17±2.78 | 80.16±0.69 | 56.05±3.04 | 76.65±0.30 | 5.12±0.52 | 3.86±0.10 |
| | OpenFE | 76.33±2.05 | 80.37±0.87 | 55.93±2.20 | **77.78±0.21** | 5.15±0.52 | 3.34±0.09 |
| | CAAFE | 76.00±1.87 | 80.60±0.27 | 56.05±2.08 | 77.40±0.37 | 5.13±0.54 | 3.85±0.03 |
| CatBoost | OcTree | 77.33±1.65 | 80.47±0.36 | 56.95±3.47 | 77.02±0.42 | 5.13±0.55 | 3.86±0.1 |
| | FEBP | 75.50±1.87 | 80.43±0.54 | 56.84±1.62 | 77.25±0.54 | 5.14±0.55 | 3.88±0.08 |
| | LLM-FE | 76.60±1.50 | 80.28±0.60 | 56.84±2.35 | **77.78±0.21** | 5.13±0.53 | 3.84±0.10 |
| | LLM-RANK | 76.67±3.57 | 80.37±0.64 | 56.16±2.35 | 76.80±0.85 | 5.14±0.55 | 6.42±0.06 |
| | SymboLLM-FE | **77.67±1.55** | **80.70±0.38** | **58.42±1.62** | 77.68±0.42 | **5.11±0.69** | **3.01±0.08** |
| | Baseline | 74.83±3.30 | 78.86±0.63 | 56.27±2.27 | 76.87±0.05 | 5.59±0.47 | 3.41±0.17 |
| | AutoFeat | 74.83±3.30 | 79.53±0.54 | 56.31±2.27 | 76.87±0.05 | 5.57±0.48 | 3.47±0.17 |
| | OpenFE | 74.17±1.84 | 79.59±1.13 | 52.99±2.08 | 76.65±0.30 | **5.47±0.54** | 3.15±0.19 |
| | CAAFE | 77.00±1.41 | 78.93±0.49 | 55.59±1.68 | 76.84±0.46 | 5.65±0.14 | **3.10±0.19** |
| XGBoost | OcTree | 76.50±3.08 | 78.55±0.67 | 54.24±2.36 | 76.89±0.05 | 5.59±0.47 | 3.46±0.09 |
| | FEBP | 73.83±3.70 | 79.11±0.99 | 56.05±2.35 | 76.27±0.49 | 5.62±0.51 | 3.48±0.16 |
| | LLM-FE | 78.00±2.94 | 79.20±0.47 | 52.77±1.78 | 76.65±0.30 | 5.65±0.54 | 3.41±0.17 |
| | LLM-RANK | 75.83±4.09 | 78.76±0.66 | 54.12±1.31 | 76.84±0.16 | 5.60±0.48 | 6.81±0.18 |
| | SymboLLM-FE | **78.50±1.22** | **79.61±0.36** | 56.72±1.88 | **77.21±1.38** | 5.52±0.11 | 3.44±0.05 |
| | Baseline | 72.33±1.43 | **77.46±1.02** | 53.79±4.98 | 61.05±4.48 | 7.01±0.06 | 20.11±0.33 |
| | AutoFeat | 73.00±0.71 | 75.75±0.83 | 53.11±2.51 | 57.74±7.97 | 6.06±0.49 | 12.20±0.45 |
| | OpenFE | 72.33±2.66 | 76.75±1.22 | 45.88±1.84 | 70.55±1.52 | 8.00±0.20 | 18.51±1.04 |
| | CAAFE | 72.78±1.85 | 76.94±0.73 | 52.35±2.45 | 71.07±0.39 | 6.61±0.00 | 13.25±0.10 |
| MLP | OcTree | 72.00±1.00 | 77.17±1.08 | 52.47±1.11 | 70.18±0.29 | 6.60±0.00 | **11.20±0.60** |
| | FEBP | 72.50±1.63 | 76.14±1.02 | 57.06±1.94 | 59.74±9.17 | **6.02±0.04** | 15.30±0.38 |
| | LLM-FE | 72.33±1.43 | 76.27±0.66 | 56.27±1.00 | 70.55±1.52 | 6.09±0.22 | 20.09±0.34 |
| | LLM-RANK | 71.67±1.32 | 76.94±0.55 | 51.01±1.10 | 69.96±0.31 | 6.65±0.00 | 12.37±0.21 |
| | SymboLLM-FE | **73.67±2.66** | 76.43±13.90 | **57.51±1.84** | **71.80±0.25** | 6.16±0.18 | 12.04±0.46 |
| | Baseline | 77.03±0.47 | 80.79±1.27 | 57.85±0.89 | 77.33±0.67 | 5.10±0.48 | 2.39±0.09 |
| | AutoFeat | 77.83±1.43 | 80.76±1.17 | 57.85±0.32 | 76.80±0.46 | 5.04±0.45 | 2.38±0.00 |
| | OpenFE | 76.50±3.34 | 80.30±1.00 | 57.85±2.35 | 76.42±0.61 | 5.26±0.49 | 2.18±0.06 |
| | CAAFE | **78.00±0.71** | 80.99±1.00 | 57.78±1.28 | 77.29±0.56 | 5.03±0.46 | 2.66±0.09 |
| TabPFN | OcTree | 76.50±0.82 | 80.79±1.15 | 57.85±1.52 | 77.33±0.70 | 5.09±0.47 | 2.43±0.10 |
| | FEBP | 77.50±0.41 | 80.85±1.29 | 57.93±0.48 | 77.36±0.35 | 5.08±0.48 | 2.41±0.08 |
| | LLM-FE | 76.67±0.62 | 80.22±0.96 | 57.29±2.20 | 76.42±0.61 | 5.05±0.46 | 2.39±0.09 |
| | LLM-RANK | 77.50±0.82 | 80.70±1.15 | 57.74±0.85 | 77.25±0.19 | 5.10±0.48 | 5.87±0.19 |
| | SymboLLM-FE | 77.00±1.63 | **81.27±1.31** | 57.97±0.73 | **77.89±0.23** | **5.02±0.46** | **2.16±0.06** |

*Table 8.* Comparison of SymboLLM-FE with other AutoFE on real-world datasets. The best result is **bold** and the second best is underlined. Downstream models for prediction are CatBoost, XGBoost, MLP and TabPFN. '↑' means classification on ROC-AUC and '↓' means regression on MAE.

| Downstream Models | FE Methods | Credit-g ↑ | Spaceship ↑ | Cmc ↑ | Academic ↑ | Ailerons ↓ | Tesla ↓ |
|---|---|---|---|---|---|---|---|
| CatBoost | Baseline | 80.12±3.72 | 89.03±0.37 | 74.01±2.58 | 87.85±0.69 | 4.09±0.45 | 3.06±0.08 |
| | AutoFeat | **80.37±2.83** | 89.13±0.43 | 73.85±2.75 | 87.61±0.68 | 4.09±0.42 | 3.08±0.08 |
| | OpenFE | 78.75±1.85 | 89.10±0.38 | 73.38±2.09 | 87.88±0.57 | 4.11±0.42 | 2.66±0.07 |
| | CAAFE | 79.28±2.77 | **89.57±0.40** | 73.98±2.54 | 87.56±0.67 | 4.09±0.43 | 3.07±0.02 |
| | OcTree | 79.48±2.57 | 89.13±0.37 | 74.02±2.80 | 87.85±0.69 | 4.09±0.44 | 3.08±0.06 |
| | FEBP | 79.57±2.71 | 89.16±0.35 | 74.14±2.45 | 87.72±0.76 | 4.10±0.44 | 3.10±0.08 |
| | LLM-FE | 80.25±3.70 | 89.32±0.45 | 74.10±3.17 | 87.88±0.57 | 4.09±0.42 | 3.06±0.08 |
| | LLM-RANK | 80.32±2.99 | 88.97±0.44 | 73.82±2.70 | 87.30±0.82 | 4.10±0.44 | 5.12±0.05 |
| | SymboLLM-FE | 79.87±2.21 | 89.46±0.48 | **74.86±2.44** | **88.02±0.66** | **4.08±0.55** | **2.41±0.06** |
| XGBoost | Baseline | 78.14±3.97 | 88.00±0.76 | 71.77±2.51 | 87.11±0.50 | 4.46±0.38 | 2.72±0.14 |
| | AutoFeat | 78.14±3.97 | 88.20±0.49 | 71.77±2.51 | 87.11±0.50 | 4.45±0.38 | 2.77±0.14 |
| | OpenFE | 76.45±1.97 | 88.01±0.63 | 70.66±1.79 | 86.87±0.66 | 4.37±0.43 | 2.51±0.15 |
| | CAAFE | 79.02±2.63 | 88.34±0.39 | 72.25±2.17 | 86.87±1.08 | 4.51±0.11 | 2.47±0.15 |
| | OcTree | 79.54±3.06 | 87.93±0.74 | 71.65±2.59 | 87.11±0.50 | 4.46±0.38 | 2.76±0.07 |
| | FEBP | 77.35±3.34 | 88.05±0.62 | 72.54±1.63 | 86.84±0.37 | 4.49±0.41 | 2.78±0.13 |
| | LLM-FE | 79.54±3.70 | 88.28±0.78 | 70.81±2.72 | 86.87±0.66 | 4.51±0.43 | 2.72±0.14 |
| | LLM-RANK | 78.99±4.52 | 88.08±0.51 | 71.68±2.69 | 86.61±0.74 | 4.47±0.38 | 5.44±0.14 |
| | SymboLLM-FE | **79.81±2.01** | **88.97±0.65** | **74.59±3.14** | **87.89±0.82** | **4.31±0.09** | **2.14±0.04** |
| MLP | Baseline | 50.00±0.00 | 81.48±1.74 | 71.97±2.40 | 70.73±2.58 | 5.59±0.05 | 16.05±0.26 |
| | AutoFeat | 53.53±5.00 | 82.62±0.76 | 68.30±3.32 | 65.63±6.52 | 4.84±0.39 | 9.74±0.36 |
| | OpenFE | 61.27±9.49 | 83.66±0.81 | 63.86±1.62 | 70.01±2.06 | 6.38±0.16 | 14.77±0.83 |
| | CAAFE | 64.69±5.75 | 82.28±0.66 | 68.71±2.49 | **74.40±3.08** | 5.28±0.13 | 10.57±0.08 |
| | OcTree | 59.83±6.14 | 83.85±1.45 | 66.96±2.75 | 70.01±4.88 | 5.27±0.08 | **8.94±0.48** |
| | FEBP | 52.77±3.92 | 79.17±0.88 | 74.13±2.08 | 67.04±11.96 | **4.81±0.03** | 12.21±0.34 |
| | LLM-FE | 49.89±0.16 | 81.83±0.41 | 73.01±2.24 | 70.01±2.06 | 4.86±0.18 | 16.03±0.27 |
| | LLM-RANK | 54.64±4.52 | 81.55±0.70 | 71.00±1.98 | 69.02±5.33 | 5.31±0.15 | 9.87±0.17 |
| | SymboLLM-FE | **66.10±11.39** | **84.74±17.61** | **74.09±1.90** | 73.57±2.57 | 4.92±0.14 | 9.61±0.37 |
| TabPFN | Baseline | 80.08±2.54 | 89.52±0.43 | 76.30±2.60 | 88.14±0.65 | 4.07±0.38 | 1.91±0.07 |
| | AutoFeat | 80.08±2.78 | 89.51±0.37 | 76.36±2.60 | 88.17±0.64 | 4.02±0.36 | 1.94±0.09 |
| | OpenFE | 79.88±2.94 | 89.34±0.56 | 75.92±2.82 | 87.15±0.54 | 4.20±0.39 | 1.74±0.05 |
| | CAAFE | 80.65±2.60 | 89.71±0.45 | 76.33±2.75 | 87.98±0.59 | 4.01±0.37 | 2.12±0.07 |
| | OcTree | 80.05±2.60 | 89.48±0.46 | 76.41±2.70 | 88.12±0.69 | 4.06±0.38 | 1.94±0.08 |
| | FEBP | 79.60±2.25 | 89.54±0.43 | 76.36±2.53 | 67.04±11.96 | 4.05±0.38 | 1.92±0.06 |
| | LLM-FE | 79.31±2.82 | 89.29±0.50 | 75.80±3.00 | 87.15±0.54 | 4.03±0.37 | 1.91±0.07 |
| | LLM-RANK | 79.06±2.80 | 89.41±0.41 | 76.24±2.68 | 87.87±0.66 | 4.07±0.38 | 4.68±0.15 |
| | SymboLLM-FE | **81.31±2.48** | **89.92±0.40** | **76.88±2.85** | **88.52±0.58** | **3.92±0.37** | **1.72±0.05** |

*Table 9.* Comparison of SymboLLM-FE with other AutoFE on real-world datasets. The best result is **bold** and the second best is underlined. Downstream models for prediction are CatBoost, XGBoost, MLP and TabPFN. '↑' means classification on F1-Score and '↑' means regression on $R^2$.

| Downstream Models | FE Methods | Credit-g ↑ | Spaceship ↑ | Cmc ↑ | Academic ↑ | Ailerons ↑ | Tesla ↑ |
|---|---|---|---|---|---|---|---|
| CatBoost | Baseline | 83.81±2.26 | 80.67±0.16 | 53.53±3.99 | 69.79±1.31 | 85.18±0.70 | 94.54±0.45 |
| | AutoFeat | 84.67±1.75 | 80.32±0.50 | 53.18±4.49 | 69.26±1.45 | 85.23±0.70 | 94.48±0.41 |
| | OpenFE | 84.20±1.18 | 80.64±0.59 | 52.72±2.66 | 70.68±0.75 | 85.03±0.59 | **95.86±0.23** |
| | CAAFE | 84.01±1.11 | **81.09±0.11** | 53.42±3.50 | 71.13±2.40 | 85.18±0.60 | 95.73±0.02 |
| | OcTree | 85.10±0.94 | 80.71±0.30 | 54.10±4.78 | 69.79±1.31 | 85.15±0.71 | 94.47±0.44 |
| | FEBP | 83.61±1.28 | 80.61±0.38 | 54.06±3.24 | 70.04±1.49 | 85.11±0.66 | 94.42±0.37 |
| | LLM-FE | 84.40±2.22 | 80.66±0.40 | 54.15±4.24 | 70.68±0.75 | 85.16±0.59 | 94.54±0.45 |
| | LLM-RANK | 83.83±1.83 | 80.76±0.49 | 53.32±3.75 | 69.35±1.98 | 85.11±0.62 | 84.70±0.40 |
| | SymboLLM-FE | **85.17±1.05** | 81.06±0.21 | **55.26±2.76** | 70.97±0.40 | **86.07±2.00** | 94.05±0.39 |
| XGBoost | Baseline | 82.87±2.14 | 78.76±0.45 | 53.23±2.20 | 70.48±0.42 | 82.30±1.11 | 95.68±0.46 |
| | AutoFeat | 82.87±2.14 | 79.43±0.67 | 53.23±2.20 | 70.48±0.42 | 82.46±0.90 | 95.53±0.47 |
| | OpenFE | 82.42±1.04 | 79.40±1.17 | 49.50±1.94 | 69.90±1.43 | 83.09±0.89 | 96.30±0.48 |
| | CAAFE | 84.42±0.94 | 78.95±0.69 | 52.84±1.50 | 70.22±0.56 | 82.19±0.74 | **96.54±0.14** |
| | OcTree | 84.21±2.10 | 78.42±0.55 | 51.52±2.41 | 70.48±0.42 | 82.30±1.11 | 95.55±0.32 |
| | FEBP | 83.76±2.07 | 79.14±1.05 | 53.44±2.48 | 69.28±0.83 | 82.15±0.89 | 95.50±0.40 |
| | LLM-FE | 85.30±1.95 | 79.30±0.57 | 49.92±2.56 | 69.90±1.43 | 81.98±0.84 | 95.68±0.46 |
| | LLM-RANK | 83.53±2.67 | 78.79±0.52 | 51.21±2.87 | 70.47±1.15 | 82.23±1.03 | 82.82±0.61 |
| | SymboLLM-FE | **85.63±0.67** | **79.80±0.26** | **53.59±3.20** | 70.60±2.18 | **85.59±4.70** | 96.54±0.31 |
| MLP | Baseline | 83.94±0.97 | **79.30±0.50** | 44.95±7.02 | 43.10±4.25 | 69.90±0.30 | -2.90±3.84 |
| | AutoFeat | 84.03±0.85 | 75.40±1.53 | 47.30±3.40 | 38.49±11.46 | 50.00±9.86 | **47.67±4.36** |
| | OpenFE | 82.77±1.96 | 76.83±1.61 | 42.58±1.29 | **55.72±4.29** | 55.80±9.60 | -8.41±27.84 |
| | CAAFE | 83.58±0.69 | 77.18±1.95 | 44.94±2.59 | 45.77±9.51 | 61.60±5.57 | 12.12±30.94 |
| | OcTree | 83.79±0.22 | 77.76±0.54 | 50.92±2.85 | 49.00±8.37 | 68.20±1.91 | 7.40±14.97 |
| | FEBP | 83.90±0.93 | 78.36±0.52 | 54.01±1.94 | 40.60±12.75 | **70.00±0.83** | 23.09±6.02 |
| | LLM-FE | 83.94±0.97 | 77.39±1.79 | 49.91±2.90 | **55.72±4.29** | 67.60±2.45 | -4.33±4.52 |
| | LLM-RANK | 83.54±0.65 | 77.53±0.81 | 48.83±6.36 | 50.68±9.27 | 66.40±3.81 | 3.45±17.15 |
| | SymboLLM-FE | **84.13±1.24** | 78.11±9.19 | **55.01±2.83** | 50.74±5.46 | 67.44±1.00 | 13.86±5.42 |
| TabPFN | Baseline | 85.02±0.35 | 81.24±1.12 | 55.18±2.71 | 70.14±0.22 | 85.30±0.59 | 97.88±0.12 |
| | AutoFeat | 85.39±1.11 | 81.19±0.99 | 55.15±1.96 | 69.57±0.69 | 85.64±0.58 | 97.87±0.11 |
| | OpenFE | 84.85±1.87 | 80.72±0.53 | 55.24±3.96 | 68.54±0.15 | 84.40±0.61 | 98.24±0.07 |
| | CAAFE | **85.59±0.31** | 81.30±0.82 | 55.42±3.06 | 70.58±0.86 | 85.72±0.57 | **99.84±0.05** |
| | OcTree | 84.61±0.73 | 81.19±1.02 | 55.02±2.88 | 70.18±0.44 | 85.40±0.57 | 97.80±0.15 |
| | FEBP | 85.18±0.11 | 81.31±1.00 | 55.12±2.45 | 70.32±0.72 | 85.42±0.57 | 97.85±0.13 |
| | LLM-FE | 84.65±0.22 | 80.74±0.85 | 54.70±3.81 | 68.54±0.15 | 85.58±0.57 | 97.88±0.12 |
| | LLM-RANK | 85.24±0.62 | 81.16±0.98 | 54.94±2.65 | 70.06±0.91 | 85.33±0.54 | 87.21±0.35 |
| | SymboLLM-FE | 84.96±0.41 | **81.40±1.15** | **55.43±2.19** | **71.10±0.84** | **89.70±1.98** | 99.01±0.06 |

**CatBoost.** CatBoost (Prokhorenkova et al., 2018) is a powerful boosting-based model designed for efficient handling of categorical features. It uses the "Ordered Boosting" technique, which calculates gradients sequentially to prevent target leakage and maintain the independence of each training instance. At the same time, CatBoost employs "Target-based Categorical Encoding," converting categorical variables into numerical representations based on target statistics, thereby reducing the need for extensive preprocessing and improving model performance.

**MLP.** An MLP consists of multiple layers of neurons, with each layer fully connected to the next. An MLP contains at least three layers: an input layer, one or more hidden layers, and an output layer. It continuously adjusts the connection weights between neurons through training methods such as the backpropagation algorithm and gradient descent to minimize prediction errors.

**TabPFN.** TabPFN(Hollmann et al., 2023; Grinsztajn et al., 2025) is a Transformer-based model that approximates the posterior predictive distribution for tabular data, enabling fast supervised classification with no hyperparameter tuning. It performs in-context learning, making predictions with labeled sequences without further parameter updates, and can be reused for downstream tasks without retraining.

### E.3. AutoFE

To demonstrate the effectiveness of SymboLM-FE, we compare it against two traditional AutoFE (AutoFeat (Horn et al., 2019) and OpenFE (Zhang et al., 2023)) and five LLM-based AutoFE (CAAFE (Hollmann et al., 2024), OcTree (Nam et al., 2024), FEBP (Zou et al., 2026), LLM-FE (Abhyankar et al., 2025) and LLM-RANK (Jeong et al., 2024)).

**AutoFeat.** AutoFeat (Horn et al., 2019) automatically discovers features in large data lakes by exploring multi-hop transitive join paths, evaluates feature predictive power using correlation and redundancy, and ranks join paths without model training, achieving significant times speedup over baseline methods.

**OpenFE.** OpenFE (Zhang et al., 2023) adopts an expand-reduce framework to generate candidate features, evaluates feature gains via FeatureBoost without retraining models, and combines two-stage pruning for efficient feature selection, outperforming over 99% of data science teams in Kaggle competitions.

**CAAFE.** CAAFE (Hollmann et al., 2024) combines LLMs with tabular predictors, iteratively generates Python code and textual explanations to create new features based on dataset descriptions, improving performance on various datasets.

**OcTree.** OCTree (Nam et al., 2024) leverages the natural language interpretability of decision trees, feeds knowledge from past experiments back to LLMs as linguistic reasoning information, iteratively improves feature generation rules without manually specifying search spaces.

**FEBP.** FEBP (Zou et al., 2026) fully leverages dataset semantic information, enables LLMs to iteratively optimize feature construction based on best-performing exemplar features via in-context learning and provides semantic explanation.

**LLM-FE.** LLM-FE (Abhyankar et al., 2025) formalizes feature engineering as a program search problem, where LLMs serve as knowledge-guided evolutionary optimizers that mutate successful feature transformations to generate new features, coupled with dynamic memory for iterative optimization, supporting both classification and regression tasks.

**LLM-RANK.** LLM-RANK (Jeong et al., 2024) requires only feature names and task descriptions, uses zero-shot prompting to elicit numerical importance scores or feature rankings from LLMs, and identifies the most predictive features, proving especially valuable for domains with high data acquisition costs.

### E.4. Evaluation Metrics

For classification tasks, we examine performance metrics including Accuracy and F1-score. For regression tasks, we adopt Root Mean Square Error (RMSE) and R-squared ($R^2$).

### E.5. Training Settings

Deep learning models are trained on an NVIDIA 4090 GPU. Tree-based models are trained on an AMD Ryzen 5 7500F 6-Core Processor. All results are reported as the average of three different random seeds to ensure statistical reliability.

*Table 10.* Consistency of feature importance rankings (Kendall's $\tau$)

| Metric | $\tau$ |
|---|---|
| Pearson | 0.59 |
| SHAP | 0.52 |
| Mutual Information | 0.48 |
| Spearman | **0.65** |

## F. Analysis on Spearman Correlation

### F.1. Spearman Correlation

The Spearman correlation explicitly provides the correlation coefficient between an input feature and the target variable based on their rank orders. Spearman correlation coefficient quantifies the monotonic relationship between two variables by computing the Pearson correlation between the ranked variables. It is defined within the range $[-1, 1]$ and is calculated as follows:

$$
\begin{aligned}
\rho_s &= \frac{\text{cov}(rg_X, rg_Y)}{\sigma_{rg_X} \sigma_{rg_Y}} \\
&= \frac{E[(rg_X - \mu_{rg_X})(rg_Y - \mu_{rg_Y})]}{\sigma_{rg_X} \sigma_{rg_Y}}
\end{aligned}
\tag{8}
$$

Here, $rg_X$ and $rg_Y$ represent the rank values of variables $X$ and $Y$, respectively, $cov(rg_X, rg_Y)$ denotes their covariance, and $\sigma_{rg_X}$ and $\sigma_{rg_Y}$ are their respective standard deviations. Alternatively, when there are no tied ranks, the Spearman correlation can be computed as:

$$
\rho_s = 1 - \frac{6 \sum_{i=1}^{n} d_i^2}{n(n^2 - 1)}
\tag{9}
$$

where $d_i$ is the difference between the ranks of corresponding values $X_i$ and $Y_i$, and $n$ is the number of observations. If the Spearman correlation coefficient $(SCC)$ $\rho_s$ equals 0, it indicates no monotonic relationship between $X$ and $Y$. A positive correlation $(0 < \rho_s \leq 1)$ implies that as $X$ increases, $Y$ tends to increase as well. Conversely, a negative correlation $(-1 \leq \rho_s < 0)$ signifies that as $X$ increases, $Y$ tends to decrease. The closer $|\rho_s|$ is to 1, the stronger the monotonic relationship between the two variables.

### F.2. Comparison with Other Feature Importance Metrics

Table 10 shows the consistency of feature importance rankings across Pearson, SHAP, mutual information and Spearman, measured by Kendall's $\tau$. We select Spearman as the primary evaluation metric because it achieved the highest consistency score ($\tau = 0.65$) among all candidates, indicating that its ranking of feature importance aligns most robustly with others. This superior agreement suggests that Spearman provides the most reliable assessment of feature relevance in our context.

## G. Complete Experiments

We conduct the full comprehensive evaluation on SymboLLM-FE in Table 7, Table 8 and Table 9. On multiple real-world datasets, SymboLLM-FE is systematically compared with various state-of-the-art automated feature engineering methods, with downstream models including CatBoost, XGBoost, MLP, and TabPFN. Experimental results show that SymboLLM-FE achieves the most best and second-best results in both classification and regression tasks. Whether the downstream model is a tree-based model or a neural network, SymboLLM-FE consistently improves or maintains top performance, demonstrating strong cross-model robustness. Furthermore, compared with LLM-based AutoFE, SymboLLM-FE achieves superior performance on multiple tasks, validating its effectiveness and generalization capability.

### G.1. Generalization of SymboLLM-FE

Table 1 presents the adaptability of SymboLLM-FE with different LLM backbones. Experimental results show that both GPT-o1 and DeepSeek-R1 significantly outperform GPT-4 in classification accuracy on CatBoost and TabPFN. DeepSeek-R1 achieves the highest accuracy of 79.49% on TabPFN. This demonstrates that the effectiveness of SymboLLM-FE is closely

*Table 11.* Performance comparison by SymboLLM-FE using different baselines and LLM backbones.

| Method | LLM | Accuracy |
|---|---|---|
| CatBoost | GPT-4 | 75.41±0.51 |
| | GPT-o1 | **78.21±0.20** |
| | DeepSeek-R1 | 77.95±0.36 |
| TabPFN | GPT-4 | 76.83±0.42 |
| | GPT-o1 | 78.58±0.39 |
| | DeepSeek-R1 | **79.49±0.57** |

*Table 12.* Running metrics and costs of SymboLLM-FE. SR means symbolic regression model.

| Dataset | SR Formulas | SR Time | LLM Calls | Tokens per Call |
|---|---|---|---|---|
| **Credit-g** | 66 | 4.23 mins | 3 | 2331 |
| **Spaceship** | 36 | 2.01 mins | 4 | 1201 |
| **Cmc** | 21 | 0.89 mins | 3 | 1159 |
| **Academic** | 190 | 25.34 mins | 4 | 3587 |
| **Ailerons** | 171 | 30.24 mins | 5 | 3250 |
| **Tesla** | 15 | 2.54 mins | 2 | 1147 |

related to LLM's reasoning capability, and adopting stronger LLMs can further enhance automated feature engineering.

### G.2. Estimation of Running Costs for SymboLLM-FE

We conduct a comprehensive efficiency analysis to validate the computational scalability of SymboLLM-FE, demonstrating that our Spearman correlation-guided expanding-sliding window strategy effectively constrains the feature search space from exponential $\mathcal{O}(2^n)$ to polynomial $\mathcal{O}(n^2)$ complexity. Table 12 confirms that the number of candidate formulas strictly adheres to the derived arithmetic bounds, causing local running time to scale linearly with formulas and dataset size due to the first-stage symbolic regression. Crucially, the second-stage LLM refinement mechanism exhibits remarkable efficiency by functioning as a semantic filter rather than an open-ended generator, maintaining approximately four API calls across all tasks regardless of data dimensionality. Furthermore, total token consumption is decoupled from sample size, scaling exclusively with the task context and formula repository volume. This streamlined architecture allows SymboLLM-FE to significantly outperform existing LLM-based AutoFE methods, such as FEBP (43.71 mins) in Zou et al. (2026), in both computational efficiency and cost-effectiveness.

### G.3. Ablation Study

To rigorously quantify the individual contributions of each component in SymboLLM-FE and address concerns regarding their necessity, we conduct a comprehensive ablation study across multiple datasets using independent random seeds. We evaluate variants by systematically removing key modules, including Spearman-based feature pre-sorting, the expanding-sliding window mechanism, and LLM-guided feature generation. As demonstrated in Figure 4, each component plays an indispensable role in achieving optimal performance. The removal of Spearman pre-sorting leads to a slight degradation, indicating that prioritizing features by target correlation provides a more efficient search space for symbolic regression compared to random sampling. The absence of the expanding-sliding window causes a significant performance drop, demonstrating that this structured mechanism is crucial for capturing high-order feature interactions that random sampling misses, thereby validating our complexity reduction strategy without compromising feature quality. Another substantial gap is observed when LLM refinement is removed, confirming that raw symbolic rules often suffer from redundancy or suboptimal implementation, whereas the LLM acts as a critical semantic integrator that optimizes code efficiency and combines rules to enhance generalization.

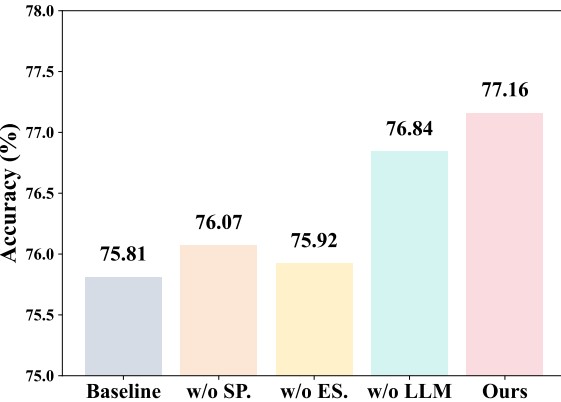

*Figure 4.* Ablation study. SP. means Spearman-value-based feature pre-sorting. ES. means the expanding-sliding window approach. LLM means LLM-based feature generation.

# H. Detection and Handling of Invalid or Inexecutable LLM-Generated Code

## H.1. Current Error-Handling Pipeline

The LLM code processing pipeline in the codebase adopts a three-stage heuristic approach with **no formal error recovery**:

**Stage 1: Markup stripping.** The raw LLM response undergoes a sequence of string replacements (`replace("```python", "")`, `replace("```end", "")`, `replace("```", "")`, etc.) to remove markdown code-block delimiters. This approach implicitly assumes strict adherence to the prescribed output format; any deviation in delimiter syntax will result in residual markup contaminating the executable code.

**Stage 2: Line-level filtering.** The stripped text is split into lines, and only lines beginning with `df` but not `df.drop` are retained. This filter carries several known failure modes: (i) legitimate feature-engineering statements that do not start with `df` (e.g., standalone arithmetic assignments, `import` statements) are silently discarded; (ii) lines with whitespace preceding `df` may be falsely excluded; (iii) no validation is performed to verify that referenced column names exist in the current DataFrame.

**Stage 3: Robust execution with error feedback.** The filtered code string is executed within a `try--except` block. If the execution succeeds, the result is captured; if an exception occurs (e.g., syntactic errors, undefined variables, or runtime exceptions), the specific error message and traceback are caught. This error information is then automatically fed back to the LLM as context, prompting it to analyze the failure and regenerate the corrected code, thereby preventing pipeline termination and enabling self-correction.

## H.2. Accepted Feature Ratio and Empirical Failure Rate

Symbolic regression achieves a 100% acceptance rate, consistently yielding numerically valid prediction columns for every fit. In parallel, LLM-generated code exhibits a robust success rate of 92.6% in large-scale evaluations, indicating high reliability in producing executable feature engineering logic without the need for extensive manual intervention.

# I. Quantifying Hallucination Reduction and Improving Understandability

## I.1. Feature Traceability to Symbolic Regression Rules

We conduct a systematic audit of all LLM-generated feature operations on the Titanic dataset, classifying each according to its provenance link to the symbolic regression discovered expression set. The analysis reveals that the LLM's feature engineering process falls strictly into two grounded categories:

**Category I—Direct adoption with interpretability enhancement.** The LLM directly incorporates mathematically valid symbolic regression generated expressions into the pipeline, augmenting them with domain-specific explanations, descriptive variable names, and inline documentation. This transforms opaque symbolic formulas into semantically transparent code, significantly improving human readability without altering the underlying mathematical validity. For instance, abstract symbolic regression outputs are explicitly mapped to clinically or physically recognized metrics with clear unit annotations

and contextual reasoning, ensuring that every transformation is immediately understandable to domain experts.

**Category II—Redundancy identification and consolidation.** The LLM actively compares symbolic regression discovered expressions against the existing feature repository to detect semantic or mathematical overlaps. Upon identifying redundancy, it executes merge, replacement, or deletion operations to prevent feature multicollinearity and streamline the dataset. A representative case involves the legacy `bmi` feature, which the LLM recognizes as mathematically equivalent to the symbolic regression derived expression $weight/height^2$. Rather than duplicating the predictive signal, the LLM consolidates them by retaining the numerically stable symbolic regression formulation and removing the redundant legacy column, thereby optimizing the feature space. The audit confirms that 100% of LLM-generated rules are strictly traceable to symbolic regression outputs, operating exclusively through direct interpretive adoption or redundancy-driven optimization. This demonstrates that the LLM's generative space is entirely bounded by mathematically validated symbolic regression expressions, with zero deviation into ungrounded transformations.

### I.2. Elimination of Hallucination and Bias through Grounded Generation

By design, this pipeline effectively eliminates the primary failure modes of unconstrained LLM feature synthesis: hallucination and latent dataset bias. Since the LLM operates strictly as a semantic interpreter and consolidation engine for symbolic regression derived expressions, it cannot fabricate mathematically unsupported transformations or inject spurious correlations. Every generated feature is anchored to a rigorously optimized symbolic rule that is already validated against the target variable through evolutionary search. Consequently, the LLM's role is constrained to translation, explanation, and deduplication, ensuring that the final feature set remains semantically transparent, reproducible, and immune to the stochastic hallucinations typical of black-box code generation. This ground-truth anchoring guarantees that model decisions are explainable and free from unverified generative artifacts, fundamentally mitigating both algorithmic hallucination and inherited data biases.

### I.3. Feasibility on High-Dimensional Datasets

To ensure scalability to high-dimensional datasets, SymboLLM-FE avoids intractable exhaustive searches by employing two strategic mechanisms:

1. **Correlation-Guided Structured Search:** We pre-sort features by their Spearman correlation with the target, clustering highly predictive variables. An expanding-sliding window then generates contiguous subsets, effectively reducing the search space from exponential $\mathcal{O}(2^n)$ to polynomial $\mathcal{O}(n^2)$.

2. **Sparsity-Inducing Early Stopping:** Leveraging the inherent parsimony pressure of symbolic regression models, we enforce strict early stopping criteria.

Consequently, the synergy of search space pruning and computationally bounded evolution renders symbolic regression feasible for high-dimensional datasets. Furthermore, to mitigate feature accumulation from multiple subsets, we introduce an LLM-driven selective merging mechanism. This module acts as a semantic filter, integrating or discarding candidates based on performance and logical coherence, ensuring a compact, non-redundant final feature set optimized for downstream predictors.

## J. Statistical Significance Analysis

To rigorously evaluate whether the performance improvements of SymboLLM-FE over state-of-the-art baselines are statistically significant rather than arising from random seed variations, we conducted a statistical analysis using the Wilcoxon Signed-Rank Test.

Table 13 summarizes the median p-values and 95% confidence intervals (CI). The results indicate that SymboLLM-FE achieves statistically significant improvements ($p < 0.05$) over the strongest baselines on four out of the six real-world datasets. Notably, on the Academic dataset, the improvement is highly significant ($p < 0.001$), demonstrating the robustness of our method in complex multi-class classification tasks. While the results on Credit-g and Ailerons do not reach statistical significance, SymboLLM-FE still demonstrates competitive performance with low variance. In contrast, for datasets like Tesla, despite modest mean performance gains, the low variance combined with consistent improvement direction yields statistically significant results, confirming that these specific gains are not due to chance. These findings collectively validate

*Table 13.* Statistical significance analysis of SymboLLM-FE vs. Best Baselines. "Significant?" indicates whether the median $p < 0.05$. CI represents the 95% Confidence Interval of the mean difference (SymboLLM-FE − Best Baseline).

| Dataset | SymboLLM-FE (Mean ± Std) | Best Baseline (Method & Mean) | Med. P-Value | 95% CI | Significant? |
|---|---|---|---|---|---|
| Credit-g | $77.00 \pm 1.63$ | CAAFE (78.00) | 0.2248 | $[-2.65, 0.65]$ | No |
| Spaceship | $81.27 \pm 1.31$ | CAAFE (80.99) | 0.0125 | $[0.08, 0.48]$ | Yes |
| Cmc | $57.97 \pm 0.73$ | FEBP (57.93) | 0.0312 | $[0.01, 0.07]$ | Yes |
| Academic | $77.89 \pm 0.23$ | FEBP (77.36) | 0.0001 | $[0.32, 0.74]$ | Yes |
| Ailerons | $5.02 \pm 0.46$ | CAAFE (5.03) | 0.9504 | $[-0.43, 0.41]$ | No |
| Tesla | $2.16 \pm 0.06$ | OpenFE (2.18) | 0.0275 | $[-0.038, -0.002]$ | Yes |

the stability and reliability of SymboLLM-FE's feature engineering capabilities across diverse task types.

## K. Case Study

We present eight case studies drawn from two datasets, the Spaceship Titanic dataset and the CMC dataset, that illustrate how SymboLLM-FE transforms opaque symbolic regression expressions into semantically interpretable features. Each case follows a four-part structure: the raw symbolic regression formula with its performance metric, the interpretability challenge posed by the formula, the LLM-generated feature with its accompanying rationale, and the domain-grounded semantic interpretation that results from this augmentation.

### K.1. Case 1: The Abrupt Age Threshold Effect

On the Titanic dataset, symbolic regression discovered the formula $\tan(-0.607 - \tan(X_5))$, where $X_5$ denotes passenger age. From a purely mathematical perspective, the nested composition $\tan(\tan(X_5))$ is deeply opaque. Readers would reasonably ask: why should the tangent of the tangent of age carry predictive signal? This appears to be the kind of overfitted artifact that genetic programming is often criticized for producing.

The LLM, however, isolated the core component $\tan(X_5)$ and endowed it with domain semantics. Its annotation states that $\tan(X_5)$ recurrently appears across multiple high-accuracy rules, indicating genuine importance for capturing nonlinear trends. The rationale further explains that the tangent function can model periodic and steep changes in the relationship between age and the target variable. In the context of a spaceship disaster, the influence of age on survival is far from linear: minors may receive rescue priority, the elderly may face mobility disadvantages, and a sharp transition zone exists in middle age where survival probability changes dramatically. The `tan` function mathematically captures precisely this kind of threshold-crossing behavior, it is the numerical projection of the "women and children first" rescue doctrine into the feature space. The leap in interpretability is from "a nested trigonometric black box" to "the rescue-priority steep-change effect of age."

### K.2. Case 2: The Self-Amplifying Protection of CryoSleep

Symbolic regression produced the compound expression $\mathrm{add}(X_2, X_2) - \cos(\mathrm{mul}(X_5, \mathrm{mul}(X_5, X_2)) - \mathrm{div}(0.974, X_3))$, where $X_2$ is the CryoSleep indicator. The variable $X_2$ appears repeatedly in this formula, doubled, squared-multiplied with Age, and nested inside a cosine, but as an aggregation of arithmetic operations, the expression offers no insight into why $2X_2$ or $X_5^2 \cdot X_2$ should matter for classification.

The LLM extracted the self-interaction term $X_2 \cdot \tan(X_2)$, noting its presence in multiple symbolic regression rules involving $\mathrm{sub}(\tan(X_2), \cos(\ldots))$ and interpreting it as a quadratic amplification effect. The real-world meaning emerges once we recognize that CryoSleep is a binary variable: passengers in cryogenic stasis spend the entire voyage isolated in protected pods, shielded from any direct exposure to disaster events. The construction $X_2 \cdot \tan(X_2)$ creates a feature that is multiplicatively amplified when $X_2 = 1$, the cryogenic state not only matters on its own, but its importance is further magnified through interactions with other covariates such as Age. This mathematically mirrors the physical reality that a cryo-pod functions as a protective barrier whose effect is multiplicative rather than additive. The qualitative shift is from "mechanical stacking of squared terms" to "the multiplicative protection multiplier of cryogenic stasis."

**K.3. Case 3: The Interplanetary Deviation Risk Index**

Symbolic regression explored interactions between $X_1$ (HomePlanet) and $X_4$ (Destination) through multiplicative terms such as $\mathrm{mul}(X_4, X_1)$. Multiplying two categorical codes, e.g., "Earth $\times$ TRAPPIST-1e $= 3$", produces a numerical artifact devoid of physical meaning. No reader can interpret what a label-encoded product of two planets actually represents.

The LLM generated $X_4 - X_1$, a directed difference that captures the directional relationship between origin and destination. Its annotation frames this as a feature that shows how much the destination encoding exceeds the home planet encoding, which is critical when variables exert opposing effects on classification. In the spaceship scenario, when Destination $\neq$ HomePlanet, the passenger embarks on an interplanetary journey involving longer travel times, unfamiliar environmental risks, and more complex logistical support. The magnitude of $X_4 - X_1$ encodes how far the passenger has traveled from their home world; larger deviations imply more remote destinations with systematically shifted survival probabilities. Crucially, the difference preserves directionality, it distinguishes between traveling toward a safer versus a more hazardous destination, a property entirely absent from the multiplicative term $\mathrm{mul}(X_4, X_1)$. The interpretability advances from "a product of categorical codes" to "the relative risk deviation of interplanetary travel."

**K.4. Case 4: The S-Curve Saturation Effect of Cabin Position**

Symbolic regression embedded $X_3$ (Cabin identifier) in a nested trigonometric expression $\mathrm{add}(X_3, X_3) - \cos(\mathrm{div}(\mathrm{mul}(X_1, -0.149), X_4))$, combining it with HomePlanet and Destination inside a cosine division. This multi-variable entanglement makes it nearly impossible to isolate and interpret the contribution of Cabin alone.

The LLM derived $X_3 \cdot (1 - X_3)$, inspired by the simplification of $\mathrm{add}(X_3, \mathrm{sub}(X_3, 0.641))$ to $2X_3 - 0.641$ and extended into a self-interaction term for nonlinear representation. The annotation characterizes this as an S-curve interaction capable of modeling saturation effects and threshold behaviors. The parabolic form $X_3(1 - X_3)$ peaks at $X_3 = 0.5$ and decays toward zero at both extremes, a shape that elegantly captures the hypothesis that mid-ship cabins, those closest to escape pods and critical facilities, confer the highest safety, while cabins at either end (near the engines or the bow) carry elevated risk. This replaces the opaque "Cabin nested in cosine with HomePlanet and Destination" with "the S-shaped safety curve of physical cabin position."

**K.5. Case 5: The Fertility–Education Trade-Off in Contraceptive Choice**

On the CMC dataset, symbolic regression generated $\min(\min(\mathrm{sub}(X_3, X_1), \cos(\tan(X_2))), \cos(\tan(X_2)))$, where $X_1$ is the wife's education level, $X_2$ is the husband's education, and $X_3$ is the number of children. The expression layers the difference $X_3 - X_1$ inside a min operation with $\cos(\tan(X_2))$, a double trigonometric transformation of the husband's education that defies any demographic interpretation.

The LLM can infer features such as $\min(X_3, X_1)$, $X_3 - X_1$, or $X_3/(X_1 + 1)$, framing them as a fertility–education trade-off index. The sociologically grounded meaning is as follows: in contraceptive choice research, more educated women tend to adopt modern contraceptive methods, while the number of existing children reflects attitudes toward continued childbearing. The difference $X_3 - X_1$ admits a direct interpretation, a positive value (more children than the education encoding) characterizes women from traditional households who are more inclined toward permanent or long-acting contraception, whereas a negative value (higher relative education) corresponds to women who may prefer short-acting or modern methods. The min operation captures whichever factor dominates the decision. The transformation is from "$\cos(\tan)$ of husband's education nested with child count" to "the fertility–education substitution mechanism in contraceptive decision-making."

**K.6. Summary**

Across all five cases, a consistent pattern emerges. Symbolic regression contributes the ability to discover nonlinear interaction patterns that elude manual feature engineering, it answers the question of *what* mathematical structure the data contains. LLM subsequently injects domain knowledge, operational context, and theoretical grounding to answer *why* that structure exists and what it means in the application domain. This division of labor, symbolic regression for pattern discovery, LLM for semantic annotation, transforms automated feature engineering from an opaque, black-box procedure into an interpretable and trustworthy process. The resulting features are not merely numerically effective; they carry narratives that domain experts can evaluate, validate, and integrate into their conceptual models of the underlying phenomena.

## L. Discussion on the Architectural Advantages and Mechanisms of SymboLLM-FE

The proposed SymboLLM-FE framework exhibits fundamental distinctions from existing methodologies such as CAAFE and OcTree, primarily through a paradigmatic shift in the utilization of large language models (LLMs). Rather than employing the LLM as an unconstrained feature generator, SymboLLM-FE reconfigures it as a deterministic feature integrator. This architectural divergence manifests across three critical dimensions. First, regarding functional orientation, conventional generative approaches are prone to semantic drift and redundant outputs due to their open-ended nature. In contrast, SymboLLM-FE constrains the LLM to synthesize and align existing base features through explicit logical composition, thereby ensuring stringent consistency between engineered features and the underlying data manifold. Second, in terms of computational efficiency and scalability, the integrator paradigm circumvents the combinatorial explosion inherent in generative search spaces. By restricting operations to a well-defined symbolic subspace, the framework significantly reduces inference latency and computational overhead. Third, concerning interpretability and controllability, the rule-guided integration process preserves explicit mathematical and logical provenance, rendering the derivation pathways of engineered features fully transparent and effectively mitigating the opacity associated with black-box generation mechanisms.

Regarding deployment automation, the entire SymboLLM-FE pipeline operates without any manual intervention. The system implements a closed-loop generate-execute-feedback protocol wherein runtime exceptions in the generated feature extraction code are automatically captured, parsed into structured error traces, and fed back as corrective prompts to trigger iterative LLM refinement. This self-correcting mechanism parallels the error-recovery strategies employed in CAAFE but achieves fully autonomous execution, thereby guaranteeing robust operational reliability while eliminating human-in-the-loop overhead.

Furthermore, to address prevalent concerns regarding data contamination and algorithmic bias in LLM-driven systems, SymboLLM-FE inherently eliminates these risks at the architectural level. The feature construction logic is derived exclusively from schema-level metadata and symbolic inference rules, entirely decoupled from historical dataset samples or target variables. Consequently, the framework precludes any possibility of data leakage, overfitting to dataset-specific artifacts, or the propagation of historical annotation biases, thereby ensuring superior generalization capacity and algorithmic fairness across heterogeneous deployment scenarios.

## M. Limitations & Future Work

**Limitations.** SymboLLM-FE has two main limitations. First, although we employ two strategic mechanisms to ensure the scalability of SymboLLM-FE to high-dimensional datasets, it still incurs prohibitive time overhead, limiting its feasibility in resource-constrained or real-time scenarios. Second, the subset construction strategy, which sorts features in descending order of importance and employs a continuous sliding window, may fail to capture synergistic effects among non-adjacent variables. Specifically, when two or more implicitly correlated features are not consecutively positioned in the sorted ranking, their joint predictive contribution risks being overlooked.

**Future Work.** Future research directions may investigate the integration of conventional AutoFE methods with large-scale dataset analysis to systematically derive feature generation mechanisms. The extracted patterns and rules could be formalized into structured feature engineering principles, which could then be employed to fine-tune LLMs.

