# OpenReview forum: "SymboLLM-FE: LLM Accelerated Symbolic Regression for Automated Feature Engineering"
_ICML.cc/2026/Workshop/FMSD — FMSD @ ICML 2026 Poster_

### Official Review · Reviewer_PgDP · 2026-05-22
**Promising Hybrid AutoFE, Overstated Claims**

**Rating:** 5
**Confidence:** 4

**Review:**

SymboLLM-FE proposes a sensible hybrid: use symbolic regression to find mathematical feature rules, then ask an LLM to turn those rules into executable feature-engineering code and refine it through downstream feedback. The best part of the paper is this division of labor. Symbolic regression gives the LLM a constrained starting point, while code execution and validation scores keep feature generation tied to data rather than free-form speculation.

The empirical results are competitive across six tabular datasets and several downstream predictors. The method is often best or near-best, and the TabPFN results show small but useful gains on datasets such as Spaceship, Academic, Ailerons, and Tesla. The ablation also supports the expanding-sliding search as an important component.

The paper overstates some claims. The stated reduction from exponential search to the squared order is better read as heuristic pruning of the candidate space, not a full runtime guarantee, since symbolic regression and LLM calls still dominate cost. Claims about eliminating hallucinations are also too strong; execution feedback can catch broken code, but not every semantically poor or leaked feature. The inconsistent naming between SymboLLM-FE and SymboLM-FE, plus a formula-description mismatch, also hurts polish. The idea is promising, but the evidence supports “competitive and constrained,” not “solves AutoFE.”

---

### Official Review · Reviewer_n8qP · 2026-05-22

**Rating:** 7
**Confidence:** 4

**Review:**

### Summary
This paper proposes SymboLM-FE, a hybrid automated feature engineering framework that combines symbolic regression with LLM-based feature refinement for tabular prediction tasks. The method first ranks raw features by Spearman correlation, then applies an expanding-sliding window strategy to construct feature subsets and derive symbolic regression rules. These statistically grounded rules and their performance scores are then provided to an LLM, which integrates them into executable feature generation code and iteratively refines the generated features using downstream validation feedback. Experiments on six real-world datasets and four Kaggle competitions show that SymboLM-FE generally improves over traditional AutoFE and LLM-based AutoFE baselines, with additional ablations supporting the usefulness of Spearman sorting, expanding-sliding search, and LLM-based rule integration.

---

### Strengths
* The paper addresses a timely and practically important problem in tabular learning, namely how to make LLM-based feature engineering more reliable and interpretable. The core idea of using symbolic regression as a statistically grounded intermediate representation before invoking an LLM is well motivated and more principled than directly asking an LLM to invent features.
* The proposed expanding-sliding window design is a useful contribution. It provides a concrete mechanism to reduce the feature subset search space from exponential complexity to polynomial complexity while still preserving structured interactions among correlated features. This makes the framework more scalable than naive symbolic regression over all feature subsets.
* The empirical evaluation is fairly comprehensive. The paper compares against both classical AutoFE methods and recent LLM-based methods across multiple task types, datasets, and downstream predictors. The ablation study further supports that the main components each contribute to the final performance, especially the expanding-sliding window strategy.

---

### Weaknesses
* Although the paper claims improved efficiency, the total computational cost of SymboLM-FE is not fully characterized. The method still requires training symbolic regression models over many feature subsets, querying an LLM, executing generated code, and iteratively validating downstream performance. The appendix also acknowledges that symbolic regression hyperparameters are sensitive to dataset size and may require extensive tuning, which weakens the practical efficiency claim.
* The experimental gains, while generally positive, are sometimes modest and not uniformly dominant across all datasets and downstream models. For example, on several settings in the main table, SymboLM-FE is close to or slightly worse than the best baseline. The paper would be stronger with more rigorous statistical testing, clearer reporting of runtime and token cost, and an analysis of when symbolic-regression-guided LLM refinement is most beneficial.